# Androgen receptor variant shows heterogeneous expression in prostate cancer according to differentiation stage

Ada Gjyrezi [1], Giuseppe Galletti [1], Jiaren Zhang[1], Daniel Worroll [1], Michael Sigouros [2], Seaho Kim[1], Victoria Cooley[3], Karla V. Ballman[3], Allyson J. Ocean[1], Manish A. Shah [1], Joseph M. Scandura [4], Andrea Sboner[2,5], David M. Nanus[1,6], Himisha Beltran [7], Scott Tagawa [1,6] & Paraskevi Giannakakou [1,6✉]

Quantitation of androgen receptor variant (AR-V) expression in circulating tumor cells (CTCs) from patients with metastatic castration-resistant prostate cancer (mCRPC) has great potential for treatment customization. However, the absence of a uniform CTC isolation platform and consensus on an analytical assay has prevented the incorporation of these measurements in routine clinical practice. Here, we present a single-CTC sensitive digital droplet PCR (ddPCR) assay for the quantitation of the two most common AR-Vs, AR-V7, and AR-v567es, using antigen agnostic CTC enrichment. In a cohort of 29 mCRPC patients, we identify AR-V7 in 66% and AR-v567es in 52% of patients. These results are corroborated using another gene expression platform (NanoString[TM]) and by analysis of RNA-Seq data from patients with mCRPC (SU2C- PCF Dream Team). We next quantify AR-V expression in matching EpCAM-positive vs EpCAM-negative CTCs, as EpCAM-based CTC enrichment is commonly used. We identify lower AR-V prevalence in the EpCAM-positive fraction, suggesting that EpCAM-based CTC enrichment likely underestimates AR-V prevalence. Lastly, using single CTC analysis we identify enrichment for AR-v567es in patients with neuroendocrine prostate cancer (NEPC) indicating that AR-v567es may be involved in lineage plasticity, which warrants further mechanistic interrogation.

[1] Division of Hematology and Medical Oncology, Department of Medicine, Weill Cornell Medicine, New York, NY, USA. [2] Caryl and Israel Englander Institute for Precision Medicine, Weill Cornell Medicine, New York, NY, USA. [3] Department of Population Health Sciences, Division of Biostatistics, Weill Cornell Medicine, New York, NY, USA. [4] Richard T. Silver, M.D. Myeloproliferative Neoplasms Center, Division of Hematology and Medical Oncology, Department of Medicine, Weill Cornell Medicine, New York, NY, USA. [5] Institute for Computational Biomedicine, Weill Cornell Medicine, New York, NY, USA. [6] Meyer Cancer Center, Weill Cornell Medicine, New York, NY, USA. [7] Dana-Farber Cancer Institute, Boston, MA, USA. ✉email: pag2015@med.cornell.edu

Prostate cancer is the most commonly diagnosed cancer in men and the second leading cause of male cancer death in the United States. Active androgen receptor (AR) signaling is central to prostate cancer development and progression[1]. Hence, AR-pathway directed therapy is the first line of treatment for patients with advanced prostate cancer[2–4]. If not received in the front-line setting, upon progression to the castration-resistant state (CRPC), second-generation potent androgen receptor pathway inhibitors (ARPi) such as abiraterone, enzalutamide, and apalutamide, targeting androgen bio-synthesis and AR-ligand binding, respectively, are routinely used to further inhibit AR-driven cancer progression.

Nevertheless, most of the tumors eventually develop ARPi resistance due in some patients to aberrant activation of AR signaling including expression of AR splice variants (AR-Vs) which lack the ligand-binding domain and are constitutively active in the nucleus[5,6]. Two of the most investigated AR splice variants are AR-V7 and AR-v567es, which are transcriptionally active in the absence of androgens. Clinically, AR-V7 mRNA expression, measured in circulating tumor cells (CTCs) of patients with mCRPC, has been associated with enzalutamide and abiraterone resistance[5,7].

While the clinical impact of AR-V7 expression has been actively being investigated, the role of AR-v567es in treatment response and disease progression is not well elucidated. An earlier study reported that AR-v567es was expressed in 23% of mCRPC bone metastases and associated with a high nuclear AR immunostaining score, and shorter overall survival[8].

In addition, our recent results from a prospective multi-institutional clinical trial of patients with mCRPC receiving taxane chemotherapy, showed that AR-V7 and AR-v567es expression in patient CTCs, was associated with lower biochemical response rates and shorter progression free survial (PFS)[9] implicating both variants in taxane resistance. As both variants are co-expressed in patient samples where bulk tumor or the entire CTC fraction is used, the relative impact of each variant alone on disease progression and treatment response has not been yet determined. Furthermore, although AR-V7 mRNA and protein detection assays have been CLIA approved[10], testing for AR-V7 has yet to be fully incorporated in routine clinical decision making, due to the lack of consensus on a unified CTC enrichment method and analytical assay. Currently, both the FDA-cleared CTC capture (CellSearch®) and the CLIA approved CTC enrichment (AdnaTest) platforms, rely on epithelial cell adhesion molecule (EpCAM)-based capture[11,12]. However, as EpCAM can be downregulated during epithelial-to-mesenchymal transition (EMT), a biological process that precedes metastatic dissemination, EpCAM-based CTC enrichment may not capture the heterogeneous pool of CTCs[13].

Herein, we report the analytical specificity and sensitivity of the AR-V ddPCR assay and its ability to detect each transcript in single CTCs. The prevalence of AR-FL, AR-V7, and AR-v567es transcripts observed by ddPCR in CTCs from patients with mCRPC, was corroborated by different gene expression platforms, NanoString™ and RNA-Seq, each performed in independent, larger cohorts of patients. Single CTC analyses from patients with mCRPC and neuroendocrine prostate cancer (NEPC) revealed significant enrichment in AR-v567es expression in NEPC, which was confirmed in NEPC organoids, heretofore unrecognized.

## Results

### AR-V dd-PCR assay performance in vitro and clinical samples.
Here, we describe the development and application of a robust ddPCR assay that can efficiently and reproducibly quantify the expression of full length AR (AR-FL), AR-V7, and AR-v567es in CTCs isolated from the peripheral blood of patients with prostate cancer. To specifically amplify each transcript of interest, we designed primers and probes to span exon to exon junctions in order to avoid co-amplification of other variants with high sequence similarity, such as the sequence similarity observed between AR-V7 and AR-V9[14] (Fig. S1a; Table S1). The AR-V ddPCR assay analytical specificity is demonstrated in experiments where cells AR-V negative are transduced with low levels of plasmids encoding each AR transcript (AR-FL, AR-V7, and AR-v567es). In this setting, the assay detected signal only when the intended transcript was expressed (Fig. S1b). In addition, we did not detect any signal using genomic DNA as input, confirming the post-transcriptional expression of AR-Vs and the exon-to exon primer specificity (Fig. S1c).

The analytical sensitivity of the assay for each transcript is demonstrated in Fig. S2, with sensitivity down to a single prostate cancer cell spiked into one million peripheral blood mononuclear cells (PBMCs) isolated from a male healthy donor. The assay exhibited robust repeatability (intra-assay variation) and reproducibility (inter-assay variation) with <10% coefficient of variation across all biological replicates (Table S2).

Next, we evaluated the performance of the AR-V ddPCR assay in CTCs and matching PBMCs isolated from six patients with mCRPC as well as in PBMCs isolated from 10 male healthy donors. (Fig. S3a, b). We observed heterogeneous AR-V expression in CTCs while the matching PBMC fraction expressed low levels of AR-FL transcript (<2 copies/sample) as did the healthy donor PBMCs. The AR-V ddPCR assay reproducibility was further tested using CTC mRNA obtained from a different cohort of 14 patients with mCRPC. The CTC mRNA was divided in half and used as input in two different assay plates run by two independent operators on different days. We observed a nearly identical expression pattern for each AR transcript between the two runs confirming the assay robustness and reproducibility in clinical samples (Fig. S3c).

### Prevalence and quantitation of AR-Vs in CTCs from patients with mCRPC.
To determine the prevalence of each AR transcript in mCRPC, we next performed the AR-V ddPCR assay in CTCs isolated from 29 patients with mCRPC, following CD45 negative depletion. The clinical characteristics of this patient cohort are shown in Table S3. We found that AR-FL was expressed in 26/29 (90%) of patients, AR-V7 in 19/29 (66%) and AR-v567es in 15/29 (52%) with 13/29 (45%) patients expressing AR-FL, AR-V7, and AR-v567es transcripts and two subjects being negative for all three transcripts (Fig. 1 and Table 1). In terms of expression levels, AR-FL was the predominantly expressed transcript with a median and mean of 8.5 and 28.74 copies/sample, respectively, followed by AR-V7 with median and mean values of 1.3, and 5.24 respectively, and AR-v567es with median and mean values of 0.14 and 1.49, respectively (Table 1).

### AR-V prevalence and quantitation using RNA-Seq and NanoString™ expression platforms from large datasets of primary and metastatic prostate cancer patient samples.
To corroborate the AR-V expression patterns and prevalence observed in the 29 patients with mCRPC by AR-V ddPCR (Fig. 1), we assessed AR-V expression, measured by either RNA-Seq or NanoString™, in three different large cohorts of patients. First we mined the RNA-Seq data available from the TCGA[15] consisting of 505 primary prostate cancer samples and from the SU2C cohort consisting of 98 mCRPC patient samples[16,17]. After obtaining the raw data from RNA-Seq or NanoString™, we determined the expression of each transcript, by counting mapped reads across exon 7 and exon 8 junction for AR-FL; across

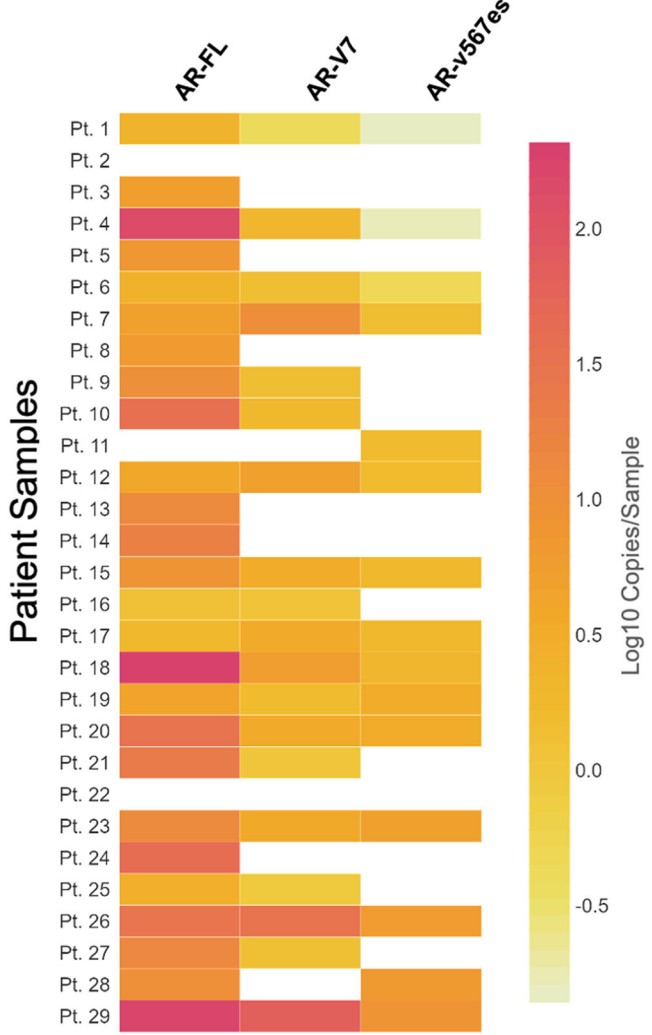

**Fig. 1 Expression of AR-FL, AR-V7, and AR-v567es in CTCs from patients with mCRPC.** CTCs were enriched from 29 patients with mCRPC by CD45 negative depletion and processed for quantification of each AR transcript by ddPCR. Heatmap of AR-FL, AR-V7, and AR-v567es expression per patient (not detected; white).

**Table 1 Transcript expression in patients with mCRPC.**

|  | AR-FL+ | AR-V7+ | AR-v567es+ |
|---|---|---|---|
| Positive Pts/Total Pts | (26/29) | (19/29) | (15/29) |
| % Positive | 90% | 66% | 52% |
| Median (copies/sample) | 8.5 | 1.3 | 0.14 |
| Mean (copies/sample) | 28.74 | 5.24 | 1.49 |
| Range (copies/sample) | 0–280 | 0–70 | 0–8.4 |

Table 1 showing the number of positive patients for each transcript, along with each respective median, mean, and range of expression values (copies/sample).

exon 3 and cryptic exon 3 junction for AR-V7; and across exon 4 and exon 8 junction for AR-v567es (Fig. S1a).

In the TCGA cohort, 98% of the primary prostate cancer samples were positive for AR-FL and 29% were positive for AR-V7. In the SU2C cohort, 97% of the castrate-resistant metastatic biopsy samples were positive for AR-FL, similar to the TCGA results, and 79% were positive for AR-V7, showing higher enrichment of this transcript in the mCRPC cohort (Fig. 2a and

**Table 2 Prevalence of AR-FL and AR-V7 in large RNA-Seq data sets from patients with primary or metastatic prostate cancer.**

| 505 primary prostate cancer samples (TCGA) | |
|---|---|
| AR-FL positive | 98% (496 in 505) samples |
| AR-V7 positive | 29% (147 in 505) samples |
| AR-FL reads range | 0–364 (mean: 55.9) |
| AR-V7 reads range | 0–72 (mean: 3.4) |

| 98 mCRPC samples (SU2C) | |
|---|---|
| AR-FL positive | 97% (95 in 98) samples |
| AR-V7 positive | 79% (77 in 98) samples |
| AR-FL reads range | 0–23321 (mean: 3771.5) |
| AR-V7 reads range | 0–852 (mean: 82.3) |

Expression of AR-FL and AR-V7 was determined in primary prostate cancer samples (TCGA, $n = 505$) and mCRPC samples (SU2C, $n = 98$) by RNA-Seq data analysis using specific exon mapping reads as described in "Methods". Transcript prevalence was calculated in both datasets and is displayed along with transcript expression range and mean.

Table 2). These results are consistent with the prevalence of AR-FL and AR-V7 transcripts in the cohort of the 29 mCRPC patients' CTCs quantified by the AR-V ddPCR assay. We also observed higher expression levels for both AR-FL (mean: 3771 reads) and AR-V7 (mean: 82 reads) in the SU2C cohort compared to the TCGA (AR-FL, mean: 56 reads; AR-V7, mean: 3 reads) (Table 2). Surprisingly, we did not detect ARv567es specific transcripts in either cohort, possibly due to very low expression levels of this variant requiring higher coverage to be detected by untargeted RNA-Seq. These results are in agreement with the report by Robinson et al.[16], where they did not detect any AR-v567es reads in the 300 of 505 TCGA patient samples, while in their expanded SU2C cohort of 125 samples they reported only 4 positives for AR-v567es.

Next, we explored AR-V expression using data obtained by a different gene expression platform, NanoString[TM]. NanoString[TM] was performed on tissues obtained from patients from our institution, including 49 from benign prostate, 89 from primary prostate cancer, and 39 from mCRPC metastases. Our targeted NanoString[TM] analysis revealed expression of all three transcripts with significantly increased expression levels with disease progression (Fig. 2b–d). Importantly, the AR-v567es transcript was expressed at much lower levels than the other two transcripts, which may explain why it was not detected by RNA sequencing, emphasizing the need for a more sensitive detection method to fully appreciate its role in prostate cancer.

**Low prevalence of AR-Vs in EpCAM^pos CTCs from patients with mCRPC.** The advantage of using CTCs is that they provide a non-invasive source of tumor cells enabling real-time interrogation in longitudinal fashion that can be used in clinical decision making. For AR-V7, an antibody-based protein expression platform, namely the Oncotype DX AR-V7 Nucleus Detect® test is CLIA approved[18]. However, for AR-V7 mRNA expression, most relevant to our platform, the only CTC CLIA approved test is the AdnaTest, which relies on epithelial cell adhesion molecule (EpCAM) expression for CTC enrichment[11]. The prevalence of AR-V7 mRNA expression in mCRPC reported by the AdnaTest is lower than what we observed in both the mCRPC patient CTCs and in the SU2C cohort. This discrepancy could be partially explained by the reported EpCAM downregulation during metastatic progression[19,20], which would limit CTC capture to the EpCAM^pos-only subtypes.

To address this concern, we enriched CTCs by CD45 negative depletion from the peripheral blood of 10 patients with mCRPC and further isolated pools of CTCs based on EpCAM expression

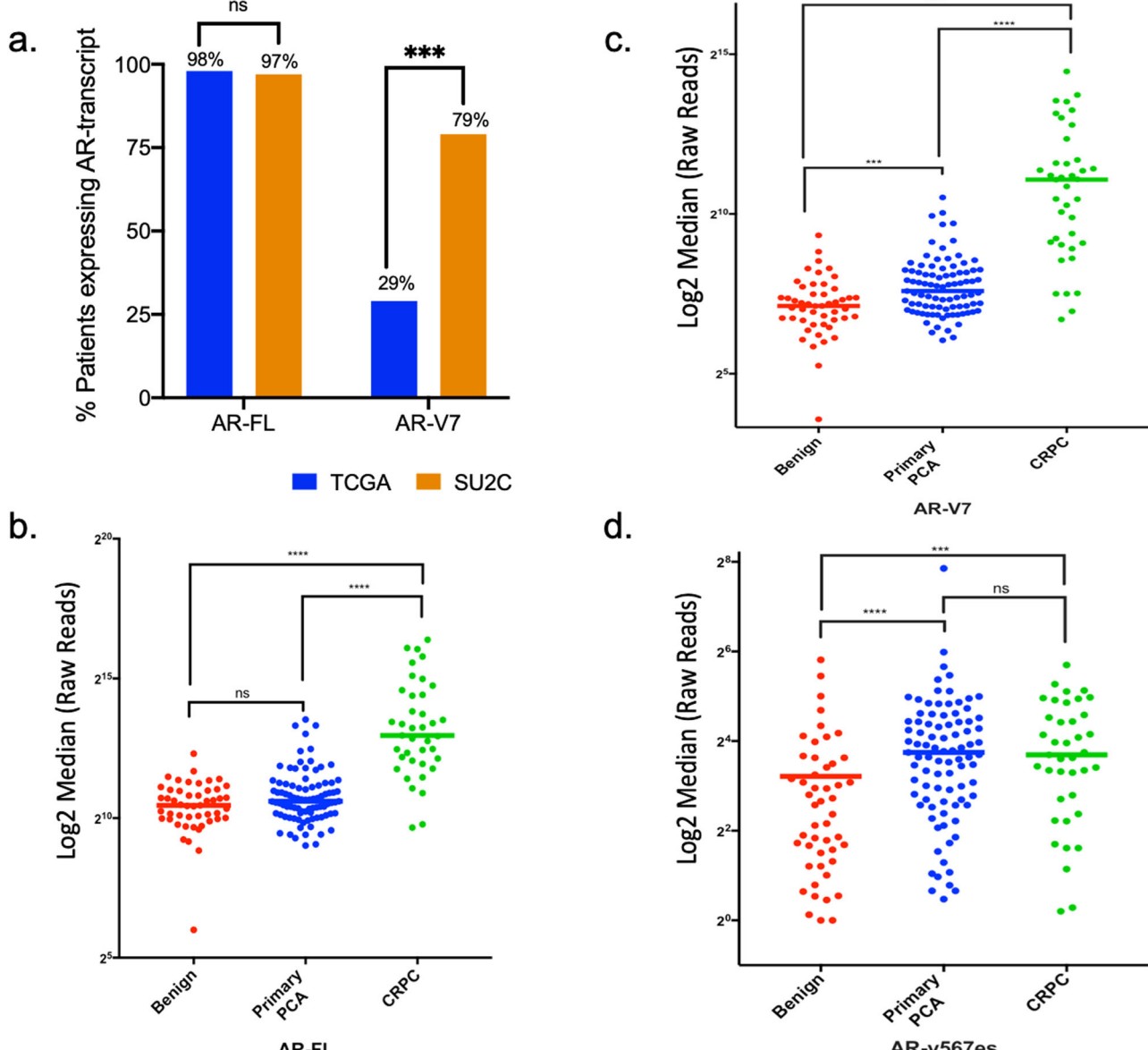

**Fig. 2 Prevalence of AR-FL, AR-V7, and AR-v567es expression in large clinical data sets from patients with localized or metastatic prostate cancer.**
**a** Expression of AR-FL and AR-V7 was determined in primary prostate cancer samples (TCGA, $n = 505$) and mCRPC samples (SU2C, $n = 98$) by RNA-Seq data analysis using specific exon mapping reads as described in "Methods". Transcript prevalence was calculated in both datasets and is displayed as % of total samples. **b–d** Expression of AR- FL, AR-V7, and AR-v567es, respectively was determined in patients with benign (red, $n = 49$), primary PCA (blue, $n = 89$), and mCRPC (green, $n = 39$) by NanoString™. Data are shown as log2 transformed raw reads. Statistical significance was determined using Mann-Whitney statistical test, ****$p ≤ 0.0001$, ***$p ≤ 0.001$, ns not significant.

using the CellCelector™ micromanipulator. We subjected these two pools of EpCAM$^{pos}$ and EpCAM$^{neg}$ CTCs to AR-V ddPCR (Fig. 3a, b). We observed no significant difference in the AR-FL prevalence between the two CTC subpopulations with AR-FL being expressed in 10/10 EpCAM$^{neg}$ and 8/10 EpCAM$^{pos}$ CTC pools. In contrast, the prevalence of both AR-V7 and AR-v567es was numerically higher in the EpCAM$^{neg}$ vs. the EpCAM$^{pos}$ pools of CTCs. AR-V7 was expressed in 9/10 (90%) EpCAM$^{neg}$ CTCs and only in 6/10 (60%) EpCAM$^{pos}$ CTCs while ARv567es was expressed in 6/10 (60%) EpCAM$^{neg}$ CTCs and only in 2/10 the EpCAM$^{pos}$ CTCs (Fig. 3b). Taken together these results indicate that CTC subpopulations have distinct AR-V expression patterns, and that antigen agnostic CTC enrichment might provide information on a more comprehensive pool of CTCs which would be missed by antigen-specific enrichment methods.

**Single CTC analysis identifies heterogeneous AR-V expression in patients with mCRPC and AR-v567es enrichment in NEPC.** Prompted by the uneven AR-V expression in EpCAM$^{pos}$ vs EpCAM$^{neg}$ CTCs and given that intra-tumoral heterogeneity is implicated in disease progression and treatment resistance, we investigated the intra-patient heterogeneity analyzing AR-V expression in single CTCs. We isolated a total of 159 single CTCs from three patients with mCRPC and subjected them to AR-V ddPCR. Each transcript alone was assessed in single CTCs from each patient. AR-FL was detected in 14/53 CTCs (26%), AR-V7 in 7/53 (13%) and AR-v567es in 6/53 (11%). While these results suggest that AR-FL is the most predominant transcript overall, when we analyzed the data on a per patient-basis we observed inter- and intra-patient heterogeneity in the proportion of single CTCs expressing each transcript, in agreement with a

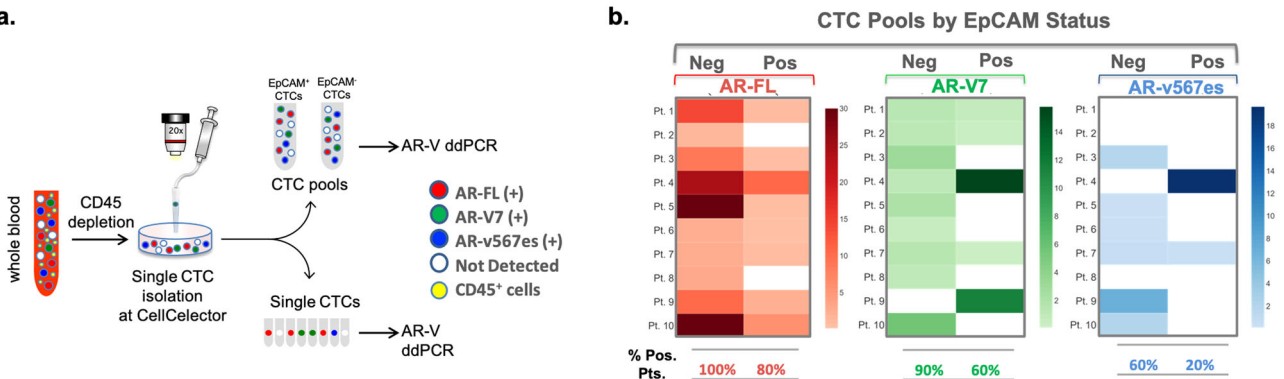

**Fig. 3 Expression of AR-FL, AR-V7, and AR-v567es in EpCAM^pos and EpCAM^neg CTC pools in patients with mCRPC. a** Experimental outline: CTCs were enriched from mCRPC patients by CD45 negative depletion followed by single-cell isolation using the CellCelector™ micromanipulator. **b** Pools of EpCAM^pos/CD45^neg or EpCAM^neg/CD45^neg CTCs were isolated from 10 patients with mCRPC and processed for quantification of each AR transcript by ddPCR. Data were normalized by the number of CTCs in each pool. Heatmap of AR-FL, AR-V7, and AR-v567es expression per patient and per EpCAM status (AR-FL, red; AR-V7, green; AR-v567es, blue; not detected, white).

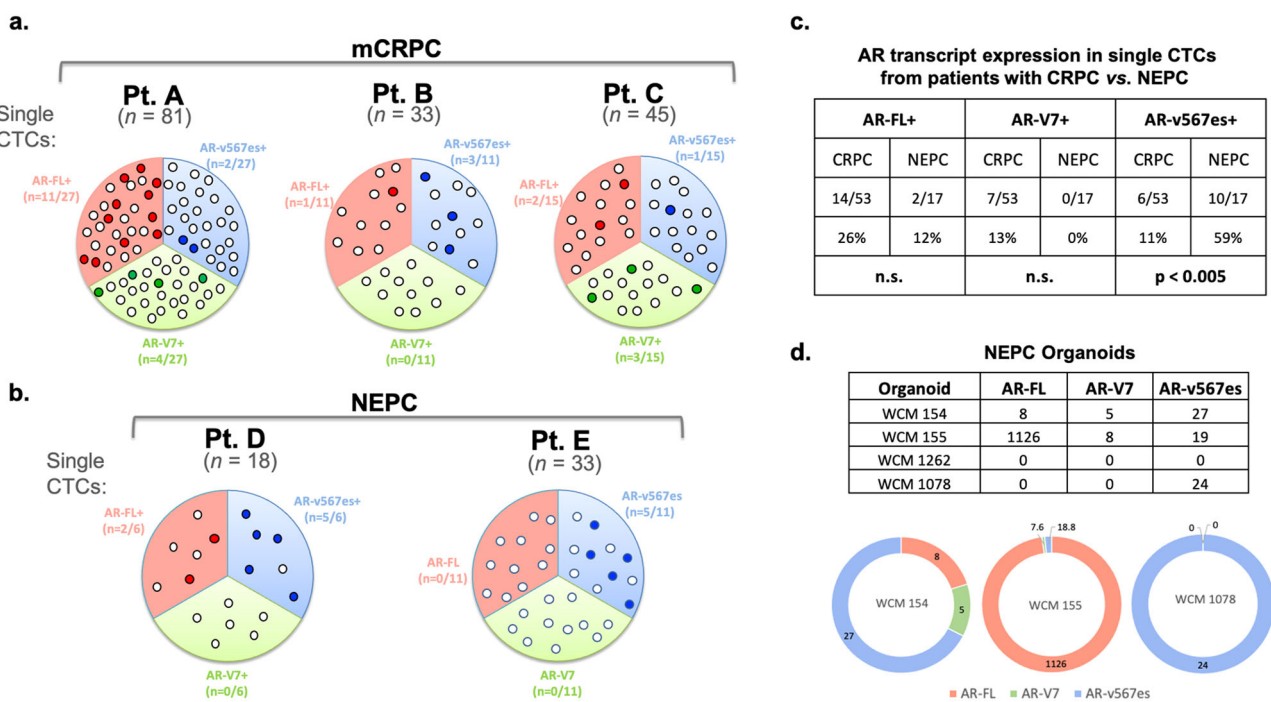

**Fig. 4 Expression of AR-FL, AR-V7, and AR-v567es in single CTCs and organoids from patients with mCRPC and NEPC. a** Single CTCs were isolated from three patients with mCRPC (A, B, C) and **b** two patients with NEPC (D, E) and processed for quantification of each AR transcript by ddPCR. Big circles represent individual patients and small circles represent single CTCs. (For patient A: we have single 81 CTCs, for patient B we have 33 CTCs, for patient C we have 45 CTCs, for patient D we have 18 CTCs and for patient E we have 33 CTCs). The total CTCs isolated from each patient were divided into three equally sized groups and each group was tested for only one of the three transcripts (For example, for patient A, we analyzed 27 single CTC for the expression of AR-FL, 27 single CTCs for the expression of AR-V7 and 27 single CTCs for the expression of AR-v567es). Colored small circles for each transcript represent positive expression, while white circles represent CTCs negative for the respective transcript. The CTCs positive for AR-FL shown in red circles; AR-V7, green circles; and AR-v567es, blue circles. **c** Prevalence of each transcript across in single CTCs from patients with CRPC vs. NEPC. Statistical significance was determined using two-tailed Fisher Exact test, n.s.; not significant for AR-FL+ and AR-V7+; $p = 0.0002$ in AR-v567es+ CTCs. **d** Table shows expression levels for each transcript in four patient-derived NEPC organoids. Data displayed in a doughnut format for three of the four organoids offer a visual display of relative transcript abundance. Transcript color coding as indicated. Data shown are copies per sample normalized to input RNA and internal loading control.

previous report on single CTC RNA-Seq from prostate cancer patients[21]. For example, patient **A** had 11 CTCs expressing AR-FL, 4 CTCs expressing AR-V7, and 2 CTCs expressing AR-v567es; making AR-FL the predominant transcript. In contrast, patient **B** had 1 CTC expressing AR-FL, 0 CTCs expressing AR-V7, and 3 CTCs expressing AR-v567es, which was the predominant transcript in this patient. Patient **C** had 2 CTCs expressing AR-FL, 3 CTCs expressing AR-V7, and 1 CTC positive for AR-v567es making AR-V7 the predominant transcript (Fig. 4a).

A similar analysis of single CTCs enriched from patients with NEPC, identified AR-v567es as the predominant transcript. Specifically, AR-FL was detected in 2/17 CTCs (12%), AR-V7 in 0/17 (0%) and AR-v567es in 10/17 (59%) (Fig. 4b). Comparison of

the AR-V expression pattern in single CTCs from patients with CRPC *vs.* NEPC, identified significant enrichment for AR-v567es transcript in NEPC (Fig. 4c, $p < 0.005$). To expand on this interesting observation, we quantified AR-V expression in organoids derived from patients with NEPC, as previously described[22]. This analysis identified AR-v567es as the predominant transcript, expressed in three of the four NEPC organoids (75%) (Fig. 4d).

## Discussion

Active AR signaling is central to prostate cancer development and progression; aberrant activation of AR signaling including expression of AR splice variants (AR-V) mediates clinical resistance to androgen deprivation therapy and to second-generation potent ARPis such as abiraterone and enzalutamide[5,18]. Currently, two of the most investigated AR splice variants are AR-V7 and AR-v567es, which are transcriptionally active in the absence of androgens. However, their clinical significance and impact on treatment decision have yet to be established. In addition, AR-V detection rates in patients with mCRPC vary widely, as do the assays used, the patient cohorts, and the source of tumor material encompassing tumor biopsies, whole blood, exosomes, and CTCs.

AR-V7 has been the most extensively investigated due to its reported association with ARPi and taxane resistance[5,7,9]. Currently, there are several assays developed to detect AR-V7 mRNA by qRT-PCR[5,8,10,23–26] or ddPCR[27–30], using liquid biopsies such as CTCs, ctRNA, exosomes, or whole blood. Interestingly, AR-V7 mRNA detection has shown higher sensitivity when CTCs are used as input, compared to other types of liquid biopsy[31]. In our AR-V assay, we chose to quantify AR-FL and AR-V expression using the ddPCR platform because it has been shown to have increased precision and sensitivity in detecting low template copies[32,33] in a complex biological background without the need for normalization, calibrator, or external references[34–36].

Interestingly, there is a wide range of AR-V7 detection rates, from 18[30] to 95%[29], even among the reports using ddPCR. In our study using CTCs from patients with mCRPC we detected AR-V7 expression in 66% of patients. This result is consistent with our earlier study where AR-V7 expression was determined in a different cohort of patients with mCRPC, using the same assay but a different CTC isolation platform (PSMA+ selection)[9]. In agreement with these results, two other studies using ddPCR for AR-V7 quantitation in CTCs from patients with mCRPC reported 50% and 53% AR-V7 detection rates[28,37]. With regards to the other ddPCR-based assays, the liquid biopsy input was either whole blood or exosomal RNA isolated from peripheral blood, and as such they cannot be directly compared to ours[27,29,30].

Notably, the primers used in our assay were designed to specifically detect AR-V7 avoiding co-amplification of the structurally similar AR-V9, which share a common nucleotide sequence in the 3′ terminal cryptic exon[14]. Some of the other assays, developed before the discovery of AR-V9, have primers and probes showing the partial overlap between AR-V7 and AR-V9[27–29,37]. The extent to which potential co-amplification of AR-V9 contributes to the discrepancy in AR-V7 detection rates remains to be determined.

For the quantification of AR-v567es mRNA from mCRPC patient samples, there are two reported assays based on qRT-PCR[8,26]. Hornberg et al. were the first to assess the expression of AR-v567es in bone metastases of mCRPC patients and reported AR-v567es expression in 23% of the cases. Liu et al. used whole blood from patients with mCRPC and reported an AR-v567es detection rate of 32%. Our assay is the first to use ddPCR with high specificity for AR-v567es detection in CTCs. We found AR-v567es expression in 52% of patients with mCRPC, albeit the expression

level was much lower than that of AR-V7 or AR-FL (Fig. 1). The high sensitivity of ddPCR could account for the higher detection rate in our cohort as compared with the more conventional qRT-PCR. Interestingly, in our earlier study where CTCs were enriched by PSMA in a different cohort of patients with mCRPC, AR-v567es was expressed in 78% of patients[9]. The distinct CTC enrichment platforms and the different lines of treatment in the two cohorts could potentially account for these discrepant results.

To address these discrepancies, we sought to determine the prevalence of AR-V7 and AR-v567es expression by analyzing large publicly available RNA sequencing data sets of primary prostate cancer (TCGA) and mCRPC (SU2C- PCF Dream Team). Our results showed that AR-V7 was expressed in 29% of patients with primary prostate cancer (TCGA) and in 78% of patients with mCRPC (SU2C- PCF Dream Team) (Fig. 2a, b). These results are in line with the 66% AR-V7 detection rate reported herein and the 67% we previously published[9]. In addition, the significant increase in AR-V7 prevalence in the metastatic pre-treated versus primary prostate cancer is consistent with published reports where AR-V7 expression is increased with disease progression[26,38]. We did not detect AR-v567es in any of the above databases, likely due to low expression levels not detectable by untargeted RNA-Seq. Using the NanoString™ platform[39], we detected AR-v567es expression in primary prostate cancer and CRPC patient samples (Fig. 2c). Similar to our ddPCR results, AR-v567es expression was much lower than that of AR-V7 and AR-FL. The overall lower expression of AR-v567es, obtained by the NanoString™ targeted sequencing could explain the challenge in detecting the transcript in the untargeted RNA sequencing databases. These results prompt the use of a more sensitive approach to clinically assess the AR-v567es prevalence in clinical samples.

In addition to the assay sensitivity, the CTC enrichment platform may also contribute to differences in AR-V detection rates. Currently, EpCAM-based methods are the most commonly used platforms for CTC enrichment, such as CellSearch® and AdnaTest. As EpCAM can be downregulated during metastasis, we decided to adopt an antigen agnostic CTC enrichment approach based on CD45 negative depletion which allowed us to capture a more comprehensive pool of CTCs. Our analysis of EpCAM^pos and EpCAM^neg mCRPC CTCs showed a higher prevalence of both AR-V7 and AR-v567es in the EpCAM^neg CTC subpopulation (Fig. 3). This observation implies that EpCAM-based enrichment captures a subset only of CTCs, which may not be representative of the entire tumor burden and disease heterogeneity, as it relates to disease progression and treatment response. Unfortunately, the absence of clinical follow-up of our patient cohort hinders any correlation of our AR-V status with clinical outcomes. Larger studies to evaluate the clinical impact of AR-Vs assessed in antigen agnostic-derived CTCs are underway.

The data obtained from the two pools of EpCAM^pos and EpCAM^neg CTCs highlighted the CTC heterogeneity in mCRPC. To further deconvolute this heterogeneity we coupled single-CTC collection with our highly sensitive AR-V ddPCR assay and observed both intra-patient and inter-patient heterogeneity of AR-V detection in single CTCs. Unexpectedly, single CTC analyses from patients with NEPC identified significant enrichment for AR-v567es expression compared to mCRPC. These findings were corroborated by the NEPC organoid analysis. NEPC is an aggressive histologic variant of PC, arising either de novo or evolving from mCRPC likely due to treatment resistance. NEPC is considered AR negative, in agreement with the immunohistochemistry results for AR protein showing that these four organoids were AR negative[22]. While we detected AR-FL mRNA signal in two of the four NEPC organoids, we cannot directly compare the detection sensitivity of the two assays, especially

since one assesses protein and the other mRNA expression. In addition, potential clonal heterogeneity within each of the organoids could account for the discrepant results. We were surprised to find AR-v567es to be the predominant transcript in NEPC, both in patient CTCs and organoids. Future studies in expanded cohorts of patients are required to validate this association as well as mechanistic studies to elucidate the role of ARv567es in NEPC development. Importantly, our data suggest that AR-v567es could serve as a potential biomarker for NEPC detection using liquid biopsies.

In conclusion, we have developed a specific and single-cell sensitive assay that can reliably detect expression of AR-FL, AR-V7, and AR-v567es transcripts from very low RNA input derived from single CTCs. By coupling single-CTC collection with the AR-variant specific ddPCR assay, we observed high intra-patient and inter-patient heterogeneity of AR variant positivity in single cells. How this diversity and range of expression correlates with disease progression and response to treatment is yet to be elucidated. Unexpectedly, we identified an imbalance in AR variant expression in EpCAM^pos versus EpCAM^neg CTCs within the same patient. These results, which require independent validation, raise concerns regarding the use of EpCAM as the most common CTC enrichment method across different platforms. Finally, the lower expression of AR-v567es in EpCAM^pos CTCs from patients with mCRPC together with its respective enrichment in NEPC suggests that this AR variant may be implicated in prostate cancer lineage plasticity.

## Methods

**Cell culture**. 22Rv1 (Cat # CRL-2505) and VCaP (Cat # CRL-2876) cells were obtained from ATCC. ATCC authenticates human cancer cell lines using short tandem repeat analysis. CWR-R1-D567 cells were kindly gifted to us from Dr. Scott Dehm (University of Minnesota)[40]. These cell lines were expanded, and early passages were frozen in liquid nitrogen. 22Rv1 and CWR-R1-D567 were maintained in RMPI1640 (Corning) supplemented with 10% FBS, 100 U/mL penicillin, and 100 μg/mL streptomycin (penicillin/streptomycin) in a 5% $CO_2$ incubator at 37 °C. VCaP cells were cultured in DMEM (Corning) with 10% FBS and penicillin/ streptomycin in a 5% $CO_2$ incubator at 37 °C.

**Plasmid transfections**. pEGFP-C1-AR-FL, pEGFP-C1-AR-V7, and pEGFP-C2-AR-567es plasmid DNA were used to exogenously express each transcript as positive controls for the development of the multiplex ddPCR assay. Six nanograms of each plasmid was transfected in AR null HEK293T cells using FuGENE® 6 Transfection Reagent according to the manufacturer's instructions (Roche, Germany). Post 24 h transfection total RNA was isolated using RNeasy Mini Kit (Qiagen, Germany, Cat. # 74104), cDNA was generated from 1 μg of total RNA using ProtoScript First Strand cDNA Synthesis (NEB, Cat. # E6300L) and cDNA samples were used for evaluating the specificity of the methodology.

**Primers and probes**. Each primer set was designed to recognize unique and distinguishable regions of AR-FL and AR variants using Primer3Plus based on the direction of the ddPCR Application Guide Bulletin 6407 (Bio-Rad). AR-FL assay recognizes unique junction between exon 7 and 8, AR-v7 assay recognizes the junction of exon 3 and cryptic exon 3, and AR-v567es assay recognizes the junction between exon 4 and exon 8. The primers were designed such at least one primer from a pair spans on the exon-exon junction to avoid unspecific amplification from other variants and synthesize from genomic DNA. The specificity of the primer design was assessed by a Nucleotide BLAST (blastn) search against the up-to-date version of the human genome reference (hg38) database and IGV (Integrative Genomics Viewer). The primers and probes were designed and purchased from Bio-Rad shown in Table S1.

**Digital droplet PCR (ddPCR)**. Droplet Digital PCR (ddPCR) is a method of absolute nucleic acid quantification based on the partitioning of a qPCR reaction sample into tens of thousands of nano-reactions (droplets) of defined volume[41,42]. After PCR, droplets that contain a template will have a fluorescent signal (positive droplets) that distinguishes them from the droplets without a template (negative droplets). All positive droplets were used to determine each transcript's expression values[28–30].

AR-FL, AR-V7, and AR-v567es transcript quantifications were carried out on a QX200 Droplet Digital PCR (ddPCR) system with automated droplet generation (Bio-Rad Laboratories). Reactions were carried out in ddPCR Plates 96-Well, Semi-Skirted (by Eppendorf). Each well contained 5.5 μl of ddPCR Supermix for Probes,

2.2 μl Reverse Transcriptase and 1.1 μl of 300 mM DTT (All components of the One- Step RT-ddPCR Advanced Kit for Probes, Cat. # 186-4022, from Bio-Rad), 1.1 μl of target-specific primers, and 11 μl of sample RNA, for a total volume of 22 μl. Immediately after droplet generation, 96-well plates containing droplet-partitioned samples were heat-sealed and PCR was carried out on a C1000 Touch Thermal Cycler (Bio-Rad) using the following cycling protocol: reverse transcription at 42 C for 60 min, enzyme activation at 95 °C for 10 min followed by 40 cycles of 95 °C for 30 s (for denaturation) and 60 °C for 60 s (for annealing/ extension), followed by a final 10 min incubation at 98 °C for enzyme deactivation. Included on each plate were the following positive and negative "plate controls": no template control (NTC), HEK 293 cells transfected with either AR-FL, AR-V7, or AR-v567es. Purified nuclease-free water was used as the negative, NTC. Transcript copy numbers for the HEK 293 cells transfected with each AR transcript or NTC can be found in Table S2.

**Reproducibility and repeatability of ddPCR**. To assess the intra-assay variability (repeatability) and inter-assay (reproducibility) of our ddPCR assay, five experimental replicates were performed with 10 ng of RNA extracted from HEK293T cells either non-transfected or transfected with plasmids encoding each respective transcript, AR-FL, ARV7, and AR-v567es. The intra-assay variability was carried out by using the same batch of RNA, pre-mix from the One Step kit, and cartridges, with samples from each repeat randomly positioned on the 96-well PCR plate. The inter-assay variability was evaluated by analyzing the same batch of RNA processed on 5 different days, with 5 different kit reagents. The standard deviation (SD) in copies/μl was calculated for each transcript. The intra- and inter-assay coefficient of variance (CV) was calculated by standard deviation/mean.

**CTC enrichment in patient samples**. Up to 20 mL of peripheral blood were collected in EDTA tubes (BD Vacutainer) from patients with mCRPC, NEPC, and healthy male subjects under Weill Cornell Institutional Review Board (IRB) approval. Each blood sample was processed within 24 h of blood draw. CTCs were enriched from the peripheral blood by depletion of CD45+ cells (RosetteSep™ Human CD45 Depletion cocktail; STEMCELL Technologies), according to the manufacturer's instructions. We also isolated matched total peripheral blood mononuclear cells (PBMCs) from 1 ml of each corresponding blood sample, by Ficoll-Paque™ PLUS (GE Healthcare Life Sciences) density gradient centrifugation. Total RNA was extracted from the enriched CTCs and matching PBMCs using the RNAeasy Plus Micro kit (Qiagen) as per the manufacturer's instructions. Samples were then processed for ddPCR analysis as described above.

**NEPC organoids**. The generation and characterization of tumor organoids derived from needle biopsies of metastatic lesions from four NEPC patients was previously described by Puca et al.[22]. RNA was extracted from each of the NEPC organoids, and the same RNA input was used for each transcript. In addition, each sample was normalized to a previously validated reference gene, GUSB as an internal loading control in the HEX channel.

**CTC micro-manipulation and single CTC selection**. The ALS CellCelector™ (Automated Lab Solutions, Germany) was utilized to automatically image, enumerate and select CTC based on immune phenotype. Briefly, after CD45 depletion, the enriched CTCs were stained live for the following cell surface antigens: EpCAM (rabbit anti EpCAM ab, clone D4K8R, Cell Signaling, 1:50 dilution) followed by anti-rabbit Alexa 568 and CD45 (mouse anti CD45-Pacific blue ab, clone H130, Biolegend, 1:50 dilution). Cells were subsequently plated on glass-bottom microwell dishes (Mat Tek Corporation) for automated scan and live imaging. CTCs were visualized under the microscope and EpCAM^pos/CD45^neg cells were isolated as single or as pools of up to 50 cells by the robotic arm of the instrument and processed for ddPCR. To identify EpCAM^neg/CD45^neg cells as bona-fide CTCs, visual inspection was performed based on established criteria including larger size, greater nuclear to cytoplasmic ratio, as well as distinct nuclear morphology[43]. For single VCaP, 22Rv1 or CWR-R1-D567 analysis, cells were plated on glass-bottom microwell dishes, and individual cells were marked under bright field before microcapillary mediated mechanical suction (Fig. S2e). Effective single-cell picking was confirmed by visual inspection of the marked positions before and after picking (Fig. S2d). Single cells were processed for ddPCR.

**RNA- Seq and NanoString™ data set analysis**. For the Cancer Genome Atlas (TCGA) and Stand Up to Cancer (SU2C) datasets, we obtained raw sequencing read data from 505 and 98 patient samples respectively, trimmed them using Trimmomatic to eliminate low quality reads and aligned to human reference genome (version hg38) using STAR. AR-FL expression was determined based on mapped reads across the junction between exon 7 and exon 8 of the AR gene. AR-V7 expression was determined based on mapped reads across the junction between exon 3 and cryptic exon 3. AR-v567es expression was determined based on mapped reads across the junction between exon 4 and exon 8.

We assessed AR- FL, AR-V7, and ARv567es mRNA expression by analyzing NanoString™ data obtained from patients at our Institution with benign prostate tissue (n = 49), primary prostate cancer (n = 89), and CRPC (n = 39). The clinical characteristics of this patient cohort have been previously published in Table 1 of

Beltran H et al.[39]. The targeted gene panel developed using the NanoString[TM] nCounter was previously described in ref. [39].

**Statistics and reproducibility**. Statistical analysis was conducted using Prism 8 (GraphPad). Specific statistical tests and parameters are indicated in figure legends. Significance was set at $P \leq 0.05$.

**Reporting summary**. Further information on research design is available in the Nature Research Reporting Summary linked to this article.

## Data availability

All data generated or analyzed during this study are included in this published article and its supplementary information files. Source data for Figs. 1–3 are provided in Supplementary Data 1. The Cancer Genome Atlas (TCGA) raw data were directly downloaded from TCGA database and Stand Up to Cancer (SU2C) raw data were directly downloaded from the database of Genotypes and Phenotypes (dbGaP). These raw datasets are available to download upon request at https://portal.gdc.cancer.gov and https://www.ncbi.nlm.nih.gov/gap/

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

## Acknowledgements

Funding for the study was provided by NCI R01 CA179100, NCI R21 CA216800, DOD Award W81XWh-19-1-0666 and partially NCI T32 CA062948 (for G.G.). The authors wish to thank Dr. Scott Dehm of the University of Minnesota for providing the CWR-R1-D567 cell line.

## Author contributions

P.G. designed the research; A.G., G.G., D.W., and S.K. performed laboratory experiments; A.G. and G.G. analyzed and interpreted the data and made the figures; J.Z., M.S., and A.S. performed the computational data analysis. V.C. and C.V.B helped with

statistical analyses of clinical outcomes; J.S. helped with the development of the ddPCR assay, A.J.O., M.A.S., D.M.N, H.B., and S.T. provision of patient samples and clinical interpretation. P.G., A.G., and G.G. wrote the paper. All authors discussed and contributed to the final manuscript.

## Competing interests

A.G., S.K., and P.G. have a pending patent application for the AR-V7 and ARv567es ddPCR assays. A.J.O. reports having served as a consultant for Celgene and Tyme. M.A.S. reports having served as a consultant/advisory board member for Merck and received research funding from Gilead, Boston Biomedical, Bristol Meyers Squib, and Merck. K.V.B. reports being a DSMB member of Takeda trial and expert witness for Johnson & Johnson, Sanofi, and Janssen Oncology. H.B. reports having served as a consultant/advisory board member for Janssen, Astella, Sanofi Genzyme, Pfizer, Astra Zeneca, Merck and has received research funding from Jansen Oncology (Inst), Abb Vie/Stemcentrx, Eli Lilly (Inst). D.M.N. reports being a DSMB member for Genentech Roche. S.T. reports receiving honorarium for consulting for Genomic Health. G.G., J.Z., D.W., M.S., J.S., V.C., and A.S. have nothing to disclose.
