## [Transparent Peer Review File · Communications Biology]

Reviewers' comments:

Reviewer #1 (Remarks to the Author):

This is a very interesting well written research paper.

Mostly I have very minor issues but one more prominent one is that, despite most Figures being really great and easy to interpret, I find the presentation of Fig 4a and 4b slightly lacking in that regard. On first visual assessment for each patient there seem to be separate cells positive (or negative) for each marker but it happens always to be the same number. The consistency of the latter fact then implies that they these actually the same cells (could be highlighted in legend!), I assume (?). With this way of presenting the data information of co-expression of AR forms in the same cells is lost. It may work better to represent CTCs once as one, slightly larger circle with division in three "pieces" coloured for the expression of the various AR forms (or uncoloured for none). Circles could be collated in columns to assemble a combined figure. I do however not insist on this change as it may in fact not work to make a better figure.

Minor issues:

1. Line 95 references should be displayed in one set of brackets
2. Line 140 replace "earlier" with "early"
3. Line 169 add "of" after "partitioning"
4. Line 191 I would not exactly call 10ng RNA input as "low" given that the method is aimed at detecting RNA from single cells expected to have roughly a thousandth of that.
5. Line 403-405 (Discussion) It may be valuable to highlight in the context of this work that AR-V7 detection in CTCs has been reported to be the most sensitive.
6. General: does not the usually extremely low level of AR-v567es suggest it is of minor importance? Apart from NEPC which has it relatively highly expressed, which is an interesting finding and could become a potential biomarker for NEPC. Should that be discussed, speculated on?

Reviewer #2 (Remarks to the Author):

In this manuscript, the authors develop a ddPCR assay for the detection of AR-V7 and AR-v567es splice variants in CTCs from patients with prostate cancer. In 29 patients with mCRPC, they identified AR-V7 in 66% and AR-v567es in 52% of patients. They then measured AR-V expression in pools of EpCAM+ vs. EpCAM- CTCs from 10 mCRPC patients and found numerically higher expression of AR-Vs in EpCAM- CTCs. Finally, they compared single CTCs from 3 patients with mCRPC compared to 2 patients with NEPC, and found significant enrichment for AR-v567es in NEPC.

The potential role of different AR splice variants (including AR-v567es) as a useful biomarker in prostate cancer is a timely topic, and there is a need to better understand the biology of different AR splice variants with respect to different cellular contexts including EMT and lineage plasticity. In this respect, this paper tackles an important problem, and in particular the observation that AR-v567es is potentially enriched in NEPC is interesting. However, weaknesses of this paper are its lack of clinical information of the patient cohorts studied, and the absence of correlation of molecular findings with clinical outcomes. There is also uncertainty with regard to the robustness of the methodology by which EpCAM negative CTCs were identified. Specific comments and questions are as follows:

1. The title is misleading because the majority of data presented in this paper is based not on single cell data, but rather on ddPCR of bulk samples. There is some single cell data, but only from 3 patients in Fig. 4 at the end of the paper.
2. Abstract, Line 60: "prevalence of both AR variants was significantly lower in the EpCAM-positive fraction." Statistical significance for this comparison was not demonstrated in the Results, and therefore the word "significantly" should not be used.

3. There is little information provided on the clinical characteristics of the patient cohorts other than that they have mCRPC or primary prostate cancer. Please provide details including PSA, disease burden, and treatment history for the cohorts subjected to CTC analysis and NanoString analysis. For patients described as having NEPC, please provide details on how this was determined (histologic appearance and/or positivity for NE markers at the protein and/or RNA levels?).
4. Figure 3: It is not clear how EpCAM negative CTCs were identified. The Methods indicate that EpCAM+ CTCs were identified for single cell picking through the use of anti-EpCAM antibody, but how were EpCAM- CTCs identified and isolated? The RosetteSep CD45+ cell depletion protocol leaves behind many contaminating CD45-neg or CD45-low hematopoietic cells such as neutrophils and eosinophils, as well as circulating endothelial cells, etc. How was it determined that the EpCAM-/CD45- cells analyzed are in fact CTCs? It is reassuring that the putative CTC pools express AR-FL, but it is surprising that they could be found without another cancer-specific marker such as PSMA. In addition, although the Discussion suggests that these EpCAM- cells are mesenchymal in nature (Line 466), there is no data provided indicating expression of any mesenchymal markers in these cells.
5. For the CTC pool experiment in Fig. 3, how was the AR and ARV expression data normalized? From the Methods, each pool contained up to 50 cells, but it is not clear how much the number of cells varied between pools. Were the data normalized by the number of cells contained in each pool?
6. Cellular-level analytical sensitivity data for the AR-v567es assay is missing. Fig. S2b shows the sensitivity data for detecting AR-FL and AR-V7 in VCaP cells spiked into PBMCs, but not for detecting AR-v567es. Cell spike-in data for sensitivity of the AR-v567es assay needs to be provided to support the claim that they “report the analytical sensitivity” of the assay (Line 123).
7. Fig. 4d: It is odd that the 2 of the 4 NEPC organoid samples express AR-FL, since most neuroendocrine prostate cancers are supposed to be AR negative. How were the data for the 4 NEPC organoid samples normalized?
8. The discussion text ends a bit abruptly. Please add a meaningful conclusion paragraph.
9. Fig. S1c: Please label the horizontal axes (droplet count). In addition, the vertical axis is cut off in the center graph.
10. Fig. S2a: Please label the horizontal axis of the graph.
11. Fig. S2d: The graphs are not labeled. Presumably one graph represents VCaP cells and the other graph represents 22Rv1 cells? In addition, was there no AR-v567es transcript present in VCaP or 22Rv1 cells?
12. Line 187 states that transcript copy numbers for each patient are in Suppl. Table 2, but the table actually shows cell line data.
13. Line 253: “AR-v56es” should be corrected to “AR-v567es”.
14. Line 286: The median expression of AR-v567es transcript in CTCs is listed as “0.14” copies per sample. However, it does not make biological sense to have an absolute RNA transcript copy number that is not an integer and is less than 1. It seems that a normalization of some sort has been performed, but this needs to be explained in greater detail.
15. Line 329: Reference 20 seems to be the wrong reference here.

Please find attached our detailed response to each of the reviewers' points. Reviewer questions are displayed in italics, for easier delineation, and our replies appear in blue colored font.

Reviewer #1 comments (Remarks to the Author):

This is a very interesting well written research paper.

Mostly I have very minor issues but one more prominent one is that, despite most Figures being really great and easy to interpret, I find the presentation of Fig 4a and 4b slightly lacking in that regard. On first visual assessment for each patient there seem to be separate cells positive (or negative) for each marker but it happens always to be the same number. The consistency of the latter fact then implies that they these actually the same cells (could be highlighted in legend!), I assume (?). With this way of presenting the data information of co-expression of AR forms in the same cells is lost. It may work better to represent CTCs once as one, slightly larger circle with division in three "pieces" coloured for the expression of the various AR forms (or uncoloured for none). Circles could be collated in columns to assemble a combined figure. I do however not insist on this change as it may in fact not work to make a better figure.

We thank the reviewer for pointing out that the description of Figure 4a and 4b is not clear and does not accurately represent the data. In the revised manuscript we have updated the figure legend to clearly describe the results. See highlighted text in Figure 4 legend.

Minor issues:

- 1. Line 95 references should be displayed in one set of brackets*
References format is now updated.
- 2. Line 140 replace "earlier" with "early"*
Updated (see yellow highlight in revised manuscript)
- 3. Line 169 add "of" after "partitioning"*
Updated
- 4. Line 191 I would not exactly call 10ng RNA input as "low" given that the method is aimed at detecting RNA from single cells expected to have roughly a thousandth of that.*
We thank the reviewer for this comment. We have clarified the wording.
- 5. Line 403-405 (Discussion) It may be valuable to highlight in the context of this work that AR-V7 detection in CTCs has been reported to be the most sensitive.*
Discussion updated.
- 6. General: does not the usually extremely low level of AR-v567es suggest it is of minor importance? Apart from NEPC which has it relatively highly expressed, which is an interesting finding and could become a potential biomarker for NEPC. Should that be discussed, speculated on?*
See updated discussion addressing these points.

Reviewer #2 (Remarks to the Author):

In this manuscript, the authors develop a ddPCR assay for the detection of AR-V7 and AR-v567es splice variants in CTCs from patients with prostate cancer. In 29 patients with mCRPC, they identified AR-V7 in

66% and AR-v567es in 52% of patients. They then measured AR-V expression in pools of EpCAM+ vs. EpCAM- CTCs from 10 mCRPC patients and found numerically higher expression of AR-Vs in EpCAM- CTCs. Finally, they compared single CTCs from 3 patients with mCRPC compared to 2 patients with NEPC, and found significant enrichment for AR-v567es in NEPC.

The potential role of different AR splice variants (including AR-v567es) as a useful biomarker in prostate cancer is a timely topic, and there is a need to better understand the biology of different AR splice variants with respect to different cellular contexts including EMT and lineage plasticity. In this respect, this paper tackles an important problem, and in particular the observation that AR-v567es is potentially enriched in NEPC is interesting. However, weaknesses of this paper are its lack of clinical information of the patient cohorts studied, and the absence of correlation of molecular findings with clinical outcomes. There is also uncertainty with regard to the robustness of the methodology by which EpCAM negative CTCs were identified. Specific comments and questions are as follows:

1. *The title is misleading because the majority of data presented in this paper is based not on single cell data, but rather on ddPCR of bulk samples. There is some single cell data, but only from 3 patients in Fig. 4 at the end of the paper.*

We thank the reviewer for this comment and agree with this assessment. We have revised the title of the manuscript to better reflect the results to “Heterogeneous AR-variant expression in prostate cancer according to differentiation stage.”

2. *Abstract, Line 60: “prevalence of both AR variants was significantly lower in the EpCAM-positive fraction.” Statistical significance for this comparison was not demonstrated in the Results, and therefore the word “significantly” should not be used.*

We have addressed the wording in the revised manuscript.

3. *There is little information provided on the clinical characteristics of the patient cohorts other than that they have mCRPC or primary prostate cancer. Please provide details including PSA, disease burden, and treatment history for the cohorts subjected to CTC analysis and NanoString analysis. For patients described as having NEPC, please provide details on how this was determined (histologic appearance and/or positivity for NE markers at the protein and/or RNA levels?).*

We thank the reviewer for this comment. The clinical characteristics of the cohort of patients subjected to CTC analyses is now included in supplementary table S3. The clinical characteristics of the cohort of patients with Nanostring™ data has been previously published, Beltran et. al. Clinical Cancer Research, 2017. We have updated the methods section to reflect this revision.

For patients with NEPC subjected to CTC analyses, NEPC status was determined by histologic appearance (tumor morphology) and conventional NE markers assessed by immunohistochemistry and reviewed by pathologist.

Excerpt from reviewer #2 overall remarks:

However, weaknesses of this paper are its lack of clinical information of the patient cohorts studied, and the absence of correlation of molecular findings with clinical outcomes.

Clinical information is now presented in Table S3. As can be seen from the table the power of meaningful clinical correlations is limited by the size and heterogeneity of the patient cohort (e.g. type and number of prior treatments and disease control status, at the time of the blood collection). A brief summary of the clinical correlation analyses is presented below (Figure I) as per reviewer’s suggestion

Fig Ia. AR-Vs correlation with response status: the status of each AR variant and its expression level was correlated with response status at the time of blood collection in the CRPC patient cohort (n=29). Wilcoxon test was performed. We observed lower expression levels of AR-FL in the patients that saw a clinical benefit (n=16) vs. the ones that progressed (n=12). While no such difference was observed between the patients with low and high levels of AR-V7 and AR-v567es transcripts.

Fig Ib. AR-Vs correlation with PSA levels: the status of each AR variant expression was also correlated with PSA levels at the time of blood collection in the same of CRPC patient cohort. No significant correlations were observed.

Fig Ic. To relate the presence of splice variants to Progression Free Survival (PFS) and Overall Survival (OS), Kaplan–Meier and Cox regression were used. No significant differences in median time to progression or median time to death (OS) were observed. These results could be attributed to the small sample size, patient heterogeneity and absence of a pre-specified blood collection time point.

Table S3. Summary of clinical characteristics

Characteristic	Value
No. of patients	29
Median age , years (range)	71 (55-90)
Median PSA, ng/mL (range)	17.6 (0.03-630.58)
Presence of bone metastases (%)	26 (89%)
Presence of visceral metastases (%)	5 (17%)
Treatment history	
No. of patients with prior therapy with ARSI (%)	14 (48%)
No. of patients with prior therapy with taxanes (%)	14 (48%)
No. of patients with prior therapy with ARSI and taxanes (%)	10 (34%)

Figure I. (For reviewer's reference only, these data are not included in the manuscript.)

Fig Ia. AR-Vs correlation with response status. A) AR-FL correlation with response status B) AR-V7 correlation with response status C) AR-v567es correlation with response status

Fig Ib. AR-Vs correlation with PSA levels. A) AR-FL pos vs. AR-FL neg, B) AR-V pos vs. AR-V7 neg, C) AR-v567es pos vs. AR-v567es neg

Figure Ic. Kaplan-Meier curve of PFS for A) AR-V7 neg vs. AR-V7 pos, B) AR-v567es neg vs. AR-v567es pos, C) any of the Vs neg vs. any of the Vs pos and Kaplan-Meier curve of OS for D) AR-V7 neg vs. AR-V7 pos, E) AR-v567es neg vs. AR-v567es pos, F) any of the Vs neg vs. any of the Vs pos.

4. Figure 3: It is not clear how EpCAM negative CTCs were identified. The Methods indicate that EpCAM+ CTCs were identified for single cell picking through the use of anti-EpCAM antibody, but

how were EpCAM- CTCs identified and isolated? The RosetteSep CD45+ cell depletion protocol leaves behind many contaminating CD45-neg or CD45-low hematopoietic cells such as neutrophils and eosinophils, as well as circulating endothelial cells, etc. How was it determined that the EpCAM-/CD45- cells analyzed are in fact CTCs? It is reassuring that the putative CTC pools express AR-FL, but it is surprising that they could be found without another cancer-specific marker such as PSMA.

We thank the reviewer for this comment. As we mentioned in the Method's section, CTCs were enriched from the peripheral blood by depletion of CD45+ cells using the Rosette Sep commercial kit. After depletion, living cells were labeled using conjugated antibodies against EpCAM, PSMA (in a few cases) and CD45. To identify EpCAM-/CD45- cells as bona-fide CTCs we performed visual inspection based on established criteria including larger size, greater nuclear to cytoplasmic ratio, as well as distinct nuclear morphology. (Fehm T, Solomayer EF, Meng S, Tucker T, Lane N, et al. (2005) Methods for isolating circulating epithelial cells and criteria for their classification as carcinoma cells. *Cytotherapy* 7: 171–185.). Importantly, in a small fraction of clinical samples we confirmed that double-negative cells identified as CTCs also expressed PSMA plasma membrane labeling. The methods section is updated to reflect the selection criteria.

In addition, although the Discussion suggests that these EpCAM- cells are mesenchymal in nature (Line 466), there is no data provided indicating expression of any mesenchymal markers in these cells.

We agree with the reviewer, this was a speculation in the discussion which we have now eliminated.

5. *For the CTC pool experiment in Fig. 3, how was the AR and ARV expression data normalized? From the Methods, each pool contained up to 50 cells, but it is not clear how much the number of cells varied between pools. Were the data normalized by the number of cells contained in each pool?*

Yes, the expression data were normalized by the number of cells present in each pool. This is now clarified in the legend of Figure 3.

6. *Cellular-level analytical sensitivity data for the AR-v567es assay is missing. Fig. S2b shows the sensitivity data for detecting AR-FL and AR-V7 in VCaP cells spiked into PBMCs, but not for detecting AR-v567es. Cell spike-in data for sensitivity of the AR-v567es assay needs to be provided to support the claim that they “report the analytical sensitivity” of the assay (Line 123).*

We thank the reviewer for the comment and have performed experiments to address this concern. We now provide spiking experiment of the CWR-R1-D567 cell line into healthy donor PBMCs. CWR-R1-D567 cell line has been engineered to express only the AR-v567es transcript as described in Nyquist et. al. PNAS 2013. These data provide proof for the specific detection of the AR-v567es transcript and the analytical sensitivity of the assay to detect this variant. The data will be included in the Supplementary Figure S2c.

7. *Fig. 4d: It is odd that the 2 of the 4 NEPC organoid samples express AR-FL, since most neuroendocrine prostate cancers are supposed to be AR negative. How were the data for the 4 NEPC organoid samples normalized?*

The data for the 4 NEPC organoid samples were normalized by 1. using the same RNA input for each transcript, and 2. by using the same housekeeping gene GUSB. This is now updated in the methods' section of the manuscript and in figure legend 4d.

8. *The discussion text ends a bit abruptly. Please add a meaningful conclusion paragraph.*

We thank the reviewer for their comment. See revised conclusion paragraph in discussion.

9. *Fig. S1c: Please label the horizontal axes (droplet count). In addition, the vertical axis is cut off in the center graph.*

We have labelled the horizontal axes in Figure S1c and corrected the vertical axis in the center graph. See updated Figure and figure legend.

10. *Fig. S2a: Please label the horizontal axis of the graph.*

We have labelled the horizontal axis.

11. *Fig. S2d: The graphs are not labeled. Presumably one graph represents VCaP cells and the other graph represents 22Rv1 cells? In addition, was there no AR-v567es transcript present in VCaP or 22Rv1 cells?*

We have added the proper labelling to each graph to display the data corresponding to each respective cell line. We also performed and included single cell data from the CWR-R1-D567 cell line which has been engineered to express only the AR-v567es transcript as described in Nyquist et. al. PNAS 2013. These data on the AR-v567es transcript quantification from single cells have been included in Figure S2e.

12. *Line 187 states that transcript copy numbers for each patient are in Suppl. Table 2, but the table actually shows cell line data.*

Thank you for bringing this typo to our attention. This is now updated in the revised manuscript.

13. *Line 253: “AR-v56es” should be corrected to “AR-v567es”.*

We have corrected the spelling as suggested in the revised version, line 259.

14. *Line 286: The median expression of AR-v567es transcript in CTCs is listed as “0.14” copies per sample. However, it does not make biological sense to have an absolute RNA transcript copy number that is not an integer and is less than 1. It seems that a normalization of some sort has been performed, but this needs to be explained in greater detail.*

In a ddPCR experiment the sample is randomly distributed into discrete droplets such that some contain no nucleic acid template and others contain one or more copies of the template. The concentration of each sample is determined by the number of positive droplets, from which the concentration is estimated by modeling a Poisson distribution. The formula used for Poisson modeling is Copies per droplet = $-\ln(1-p)$ where p = fraction of positive droplets. For our experiments, we used the QX200 Droplet Digital PCR (ddPCR) system with automated droplet generation and the QuantaSoft software commercially available from Bio-Rad Laboratories. The concentration of each target transcript in our samples was determined by the fraction of positive over total number of droplets and calculated based on their Poisson-based 95% confidence intervals. For very low input samples, target transcript concentrations of <1 copies/sample can be obtained.

15. *Line 329: Reference 20 seems to be the wrong reference here.*

We have added the correct reference: “Clinical Utility of CLIA-Grade AR-V7 Testing in Patients With Metastatic Castration-Resistant Prostate Cancer” Markowski et al JCO 2017

REVIEWERS' COMMENTS:

Reviewer #2 (Remarks to the Author):

The authors have done an excellent job responding to reviewer comments, including making appropriate revisions to their manuscript and providing additional data as needed.